# Residues and Dietary Risk Assessment of Prohexadione-Ca and Uniconazole in *Oryza sativa* L. and *Citrus reticulata* Blanco by Liquid Chromatography-Tandem Mass Spectrometry

**DOI:** 10.3390/molecules28062611

**Published:** 2023-03-13

**Authors:** Hui Ye, Yuqin Luo, Yanjie Li, Xiangyun Wang, Hongmei He, Jinhua Jiang, Jianzhong Yu, Changpeng Zhang

**Affiliations:** State Key Laboratory for Managing Biotic and Chemical Threats to the Quality and Safety of Agro-Products, Ministry of Agriculture and Rural Affairs Key Laboratory for Pesticide Residue Detection, Institute of Agro-Products Safety and Nutrition, Zhejiang Academy of Agricultural Sciences, Hangzhou 310021, China

**Keywords:** *Oryza sativa* L., citrus fruit, dietary intake risk assessment, plant growth regulators

## Abstract

A simple and sensitive method for the simultaneous quantitation of prohexadione-Ca and uniconazole in the field experiment of *Oryza sativa* L. and *Citrus reticulata* Blanco was established using solid-phase extraction (SPE) with polymer anion exchange (PAX) and Florisil followed by LC-MS/MS. The method demonstrated excellent linearity (R^2^ > 0.999 0), trueness (recoveries between 95~105%), precision (CVs between 0.8~12%), sensitivity, and repeatability (LOQ of 0.05 and 0.01 mg/kg, respectively). Residue tests were conducted in the field at 12 representative sites in China, revealing final concentrations of prohexadione-Ca and uniconazole in brown rice, rice hull, and rice straw to be below 0.05 mg/kg, while in whole citrus fruit and citrus pulp, they were below 0.01 mg/kg. These were below the maximum residue limits specified in China. The chronic dietary risks of prohexadione-Ca and uniconazole in rice crops and citrus fruits were calculated to be 0.48% and 0.91%, respectively. Our research suggests that the chronic risk associated with the daily consumption of rice crops and citrus fruit at the recommended dosage is acceptable.

## 1. Introduction

*Oryza sativa* L. (rice) and *Citrus reticulata* Blanco (citrus) are globally significant food crops and economic crops with wide distribution across hundreds of countries [1,2]. They have a rich history of cultivation and consumption in China. Milled white rice is a popular food choice among Chinese consumers. However, brown rice, which is considered a whole grain, has gained attention from those with grain intolerance due to its high dietary fiber and active substances [3]. Additionally, rice hull, rice straw, and other by-products are valuable resources that can be used for various purposes, such as livestock feed, biofuels, biocomposites, compost, and fermentation applications [4,5,6]. Citrus fruits are an essential source of nutrients for humans, providing dietary fiber, protein, vitamins, and antioxidants [7,8]. Citrus juice is the primary commodity in many countries where citrus is produced [9], while by-products such as essential oils, pies, molasses, dried pericarp, pectin, blend syrup, and others are also important components of the citrus industry chain [10,11]. Therefore, there is growing interested in rice and citrus due to their nutritional and economic significance.

Crop lodging is a common phenomenon in agricultural production of rice, wheat, and citrus. Lodging refers to the bending or breaking of crop stems due to wind, rain, or other environmental factors, leading to a reduction in the yield and quality of the crop [12,13]. Additionally, the growth of new shoots can also affect the reproductive growth of the plant, leading to a decrease in yield. Therefore, it is important to use effective methods to prevent lodging and promote the healthy growth of crops. Plant growth regulators (PGRs) are a type of pesticide used for regulating plant growth and development. They include quaternary ammonium salts (such as chlormequat chloride and mepiquat chloride), triazoles (such as paclobutrazol and uniconazole), and cyclohexane carboxylic acids (such as prohexadione calcium and trinexapac-ethyl); these PGRs work by inhibiting gibberellin biosynthesis, promoting plant photosynthesis and metabolism, and significantly increasing crop yield [14,15,16,17,18]. Currently, paclobutrazol is one of the most commonly used plant growth regulators for controlling lodging in rice [19]. However, its use can lead to issues such as residual damage and reduced yields, despite its ability to reduce plant height. The new plant growth inhibitor, prohexadione-calcium, has shown promising results in increasing rice yield while also reducing plant height and internode length at a dose of 200 mg a.i./ha [20]. Furthermore, it has been found to degrade easily in the soil through the action of microorganisms, leaving no harmful residue behind [21]. In contrast to paclobutrazol, uniconazole has been reported to exhibit a higher level of safety, greater bioactivity, and a faster rate of soil degradation [22]. The use of uniconazole increased rice stem diameter and wall thickness increased, which enhanced the flexural capacity of the plant, and significantly reduced rice plant lodging. This reinforced the effect of filling rice with inferior grain [23]. In addition, uniconazole has been found to effectively reduce the number of autumn shoots of citrus trees, prevent autumn shoots from consuming tree nutrients, and preserve tree shapes for fruit trees, thus exhibiting a shoot control effect [24]. Therefore, in agricultural production, PGRs such as prohexadione-Ca and uniconazole are used to accelerate the reproductive growth of plants to increase crop yields [12,25].

Prohexadione-Ca (calcium 3-oxido-5-oxo-4-propionylcyclohex-3-enecarboxylate) and uniconazole ((E)-1-(4-Chlorophenyl)-4,4-dimethyl-2-(1H-1,2,4-triazol-1-yl) pent-1-en-3-ol) are a type of PGRs that are widely used to regulate plant physiologic activities at low concentrations and improve product quality [26,27]. Several studies have revealed that the use of gibberellin biosynthesis inhibitors such as prohexadione-Ca and uniconazole compounds has a more significant and safer effect compared to their individual use [28,29]. The chemical structures of prohexadione-Ca and uniconazole were shown in Figure 1. Prohexadione-Ca and uniconazole have been reported to increase rice and citrus yields by regulating the nutritional and reproductive growth of these crops through foliar spraying [23,30,31]. However, the use of the triazole plant growth regulator, paclobutrazol, like uniconazole, selectively affects the community structure and diversity of soil microorganisms in the ecological environment [32]. Furthermore, excessive use of PGRs poses several potential threats to human health, such as precocious maturation, impaired reproduction, carcinogenic acute toxicity, and neurotoxicity [33,34,35]. Among them, uniconazole and other pesticides were found in 480 batches of traditional Chinese medicine samples in the traditional Chinese medicine market. The residue of Radix Ophiopogonis was found to exceed the maximum residue limit established by GB 2763-2016 in China, which could pose potential risks to the environment and human health [36,37]. However, few studies have investigated the residue and dietary risk assessment of PGRs such as prohexadione-Ca and uniconazole, especially in dairy foods such as rice and citrus. Therefore, it is urgent to verify the safety of the combination of prohexadione-Ca and uniconazole in the daily processing of products of citrus and rice.

Up to now, Choi et al. developed a method for monitoring prohexadione in Chinese cabbages and apples [38]. Zhang et al. established a method to detect uniconazole in wheat and soil by field applications [39]. There are currently no reports on the analysis method, field residue in multiple regions, or dietary intake risk of prohexadione-Ca and uniconazole in *Oryza sativa* L. and *Citrus reticulata* Blanco crops that are harvested under good agricultural practices (GAP). Therefore, this research aimed to (1) develop and validate a simple and sensitive method to the simultaneous quantitation of prohexadione-Ca and uniconazole in *Oryza sativa* L. and *Citrus reticulata* Blanco under field conditions, (2) evaluate the residue levels of these plant growth regulators in actual *Oryza sativa* L. and *Citrus reticulata* Blanco samples collected from twelve locations in China, and (3) assess the dietary risk associated with the consumption of these crops based on the observed residue levels. The results of this study will provide important information for the safe and sustainable use of these plant growth regulators in agricultural production, as well as for the assessment of the dietary safety of *Oryza sativa* L. and *Citrus reticulata* Blanco consumption.

## 2. Results and Discussion

### 2.1. Method Optimization of Prohexadione and Prohexadione-Ca

In the Chinese standard for food additives (GB 2763-2021), the residue of prohexadione-Ca is defined as prohexadione, which is still referred to as prohexadione-Ca in this study for consistency [40]. To ensure accuracy and consistency in analysis, this study established methods for quantifying both prohexadione and prohexadione-Ca. Prohexadione and prohexadione-Ca are pesticides that are water-soluble but insoluble in organic solvents. In order to efficiently extract these compounds from rice and citrus samples, a water and 0.1% formic acid/water oscillation extraction method was tested. The resulting extract was then passed through a PAX mixed anion exchange column, which could effectively adsorb the ionized cyclic acid in the extract. After the column was washed, the prohexadione and prohexadione-Ca were eluted using a 10% formic acid/water-methanol (*v*/*v*, 1:9) solution, ensuring that the eluent contained pure prohexadione and prohexadione-Ca. The method established by Choi et al. involves using 20 mL of acetonitrile for extraction, followed by rotary evaporation. After liquid-liquid distribution with 50 mL of dichloromethane and further evaporation, the sample is subjected to a series of complex processes, including the use of a Strata SAX cylinder, which requires extensive use of organic solvents and increases the cost of residue analysis [38]. The method established in this study for the analysis of prohexadione and prohexadione-Ca in agricultural products utilizes a minimal amount of reagents and involves straightforward processing steps during sample preparation, resulting in a relatively short overall processing time. As a result, the method is both cost effective and environmentally friendly, helping to minimize the risk of environmental pollution. Overall, this approach provides an efficient, economical, and sustainable means of detecting pesticide residues in agricultural products.

### 2.2. Method Validation

The trueness (accuracy), sensitivity, and precision of the developed analytical method were validated, and the important parameters of the matrix-based standard curves are summarized in Table 1. The calibration curves used to quantify the citrus tests exhibited excellent linearity over the studied range. The prohexadione curve ranged from 2 to 200 ng/mL, while the uniconazole curve ranged from 0.5 to 100 ng/mL, and both displayed correlation coefficients exceeding 0.999 0. The fortified recovery study was performed to evaluate the accuracy and precision of the developed method in quantifying prohexadione, prohexadione-Ca, and uniconazole residues in citrus and rice samples. The study involved adding known amounts of the target analytes to blank citrus and rice substrates at three different concentrations. For prohexadione and prohexadione-Ca, the concentrations were 0.05, 0.50, and 0.50 mg/kg in rice, and 0.05, 0.50, and 1.00 mg/kg in citrus. For uniconazole, the concentrations were 0.01, 0.50, and 1.00 mg/kg and 0.01, 0.50, and 1.00 mg/kg in citrus. The recovery tests for both citrus and rice were carried out in 5 parallel for each concentration. The recovery results are presented in Figure 2, where the error bar represents the coefficient of variation. The developed method showed good accuracy with average recoveries ranging from 74% to 106% for prohexadione, 75% to 106% for prohexadione-Ca, and 81% to 108% for uniconazole, with coefficients of variation ranging from 0.8% to 12% in five different substrates of rice and citrus. The limit of quantification (LOQ) for prohexadione and uniconazole in brown rice, rice hull, rice straw, whole citrus fruit, and citrus pulp was determined to be 0.01 mg/kg and 0.05 mg/kg, respectively. The sensitivity, trueness, and precision results met the requirements of analysis according to the NY/T 788-2018 guidelines [41].

When comparing the interference caused by the analyte’s solvent solution with the interference generated by the matrix effect (ME), the detector response can be affected. Therefore, it is necessary to consider matrix effects, with an ME value of >20% indicating signal reinforcement, ME < 0% representing signal inhibition, and ME values in the range of 0 to 20%, suggesting no matrix effect [42]. Therefore, to obtain more conclusive results, this study utilized standard matrix-matched calibration curves to identify residues in rice tests, while establishing standard solvent calibration curves in citrus tests for quantification using the external standard method. Appendix A provided representative chromatograms.

### 2.3. Prohexadione-Ca and Uniconazole Residues in Citrus and Rice under Field Conditions

Field residues of prohexadione-Ca and uniconazole were analyzed in rice crops and citrus fruits at twelve representative sites to determine the final residue results of these PGRs in grain and fruit samples. The results were shown in Table 2. At 12 representative sites, prohexadione-Ca and uniconazole residues were determined in brown rice, rice hull, and rice straw at rice maturity (62–112 days after application) and in whole citrus fruit and citrus pulp at maturation stage (134–191 days after application). After spraying 30% prohexadione-Ca and uniconazole water-dispersible granules on rice (62–112 d) and on citrus (134–191 d), the prohexadione-Ca and uniconazole residues in rice and citrus samples were less than 0.05 mg/kg and 0.01 mg/kg, respectively, which were either not detectable or below the LOQ and LOD established in this study. At the late stage of rice or citrus development, the majority of field residues of prohexadione-Ca and uniconazole had dissipated, with some residues being negligible. It has been reported previously that the average half-life of prohexadione-Ca and uniconazole in potato plants, wheat plants, and wheat, which belong to easily degradable pesticides, was less than 30 days [42,43]. In this experiment, prohexadione-Ca and uniconazole were applied during the growth period of rice and citrus, with a focus on the residual status in the mature period. The residues in 12 representative areas of China were found to be below the LOQ of the method established in this study, and all sampling days were more than 30 days after application. These results were consistent with prior research and reflect the pesticide characteristics and dissipation behavior of plant growth regulators such as prohexadione-Ca and uniconazole.

### 2.4. Stability of Prohexadione-Ca and Uniconazole Residues in Stored Samples

Appendix A illustrated the stability of prohexadione-Ca and uniconazole in the frozen samples of brown rice, rice hull, rice straw, whole citrus fruit, and citrus pulp. This demonstrates that the residual concentrations of prohexadione-Ca and uniconazole in the frozen samples remained above 81% of the added dosage (0.5 mg/kg) during storage at −18 °C, and their degradation rates were 30%, which meets the evaluation requirements of NY/T 3094-2017(Guidelines for Storage Stability of Pesticide Residues) [39]. Based on the study by Li Y J et al., citrus fruits are known as the typical commodity with high water and acid content [44], while the rice was characterized by its high starch content in grains [45]. Therefore, under frozen storage circumstances (<−18 °C), prohexadione-ca and uniconazole were found to be stable for up to 179 days in high starch matrices and 380 days in matrices with high water-acid content, respectively.

### 2.5. Dietary Intake Risk in Citrus and Rice

#### 2.5.1. The Maximum Residue Limit (MRL)

The maximum residue limits (MRLs) for prohexadione-Ca and uniconazole in rice and citrus were currently established by China, Japan, the United States, and European Union countries. For brown rice (rice and its products), the MRL for prohexadione-Ca was 0.05 in China and 0.2 in Japan, while the MRL for uniconazole was 0.1 in both countries. In citrus fruits, the MRL for prohexadione-Ca was 0.01 in the EU and 3.0 in the US, while that for uniconazole was 0.01 in the US and 0.3 in China. The residue analysis of prohexadione-Ca and uniconazole in rice and citrus showed that the residual value of prohexadione-Ca in citrus may exceed or be below the EU MRL because they were found to be below the LOQ established in this study. Therefore, as the field experiment was conducted in 12 representative locations in China, the study carried out further analysis using the lowest MRL established in China. As shown in Figure 3, after spraying prohexadione-Ca and uniconazole on rice and citrus at these locations, the residue levels were much lower than those of the MRLs established in China. Since PGRs are typically used during the growth period of rice and citrus, they were considered to be safer than other pesticides at the edible maturity stage.

#### 2.5.2. Dietary Risk Assessment

Residue safety assessment of MRL is traditionally regulated on an individual basis in food safety. Although the residues of prohexadione-Ca and uniconazole in rice and citrus were below the MRLs set in China, the residue levels of prohexadione-Ca in citrus cannot be accurately assessed according to the MRLs set by the EU, indicating potential risks. Therefore, building upon China’s dietary risk assessment model, this study further investigated the food safety of prohexadione-Ca and uniconazole at the individual level. The conclusion of long-term dietary risk assessments was presented in Table 3 and Table 4. Table 3 showed that STMRs of prohexadione-Ca and uniconazole in rice and citrus samples were 0.05 and 0.01 mg/kg, respectively, resulting in a total NEDI for prohexadione-Ca, and uniconazole was 0.0608 and 0.0114 mg, respectively. The ADIs of prohexadione-Ca in China were 0.2 mg/kg b.w. and of uniconazole established in China were 0.02 mg/kg b.w. [40], and the average b.w. of Chinese adults in China was 63 kg [46], hence, the highest ADIs for prohexadione-Ca and uniconazole was 12.60 mg and 1.26 mg, respectively. The calculated NEDIs of prohexadione-Ca and uniconazole in this present study were less than the maximal ADIs. After spraying, the Risk Quotient (RQ) of prohexadione-Ca and uniconazole water-dispersible granules on citrus fruits and rice crops was far less than 100%. The RQs of prohexadione-Ca and uniconazole were 0.48% and 0.91%, respectively, of the total population. Therefore, the long-term ingestion of the mild residue of prohexadione-Ca and uniconazole from use does not have an impact on public health, as the overall NEDI does not rise above the peak ADI.

## 3. Materials and Methods

### 3.1. Chemicals and Reagents

The prohexadione (99.5% purity), prohexadione-Ca (93.4% purity) and uniconazole (97.5% purity) were provided by two Chem service providers, Dr. Ehrenstorfer Gmbh and Aladdin, respectively. Methanol and acetonitrile (HPLC grade) were purchased from Merck (Rahway, NJ, USA), formic acid (HPLC-grade, Anaqua Chemicals, Houston, TX, USA), water (Watson Group, Guangzhou, China), Agela Cleanert PAX (500 mg/6 mL) and Florisil (1 g/6 mL), Agela Technologies (Tianjin, China), and Sodium chloride (Yida Chemical, Shanghai, China).

Individual stock solutions of prohexadione and prohexadione-Ca were dissolved in pure water at a concentration of 100 mg/L each. The uniconazole stock solution was prepared by dissolving it in acetonitrile at a concentration of 200 mg/L. All standard solutions are stored in the refrigerator at 4 °C in the dark when not in use.

### 3.2. Field Experiments

#### 3.2.1. *Oryza sativa* L.

When conducting residue trials of prohexadione-Ca and uniconazole in rice crops, this study followed the NY/T 788-2018 (Guideline on Pesticide Residue Trials) issued by the Ministry of Agriculture of the People’s Republic of China [41]. The field experiments were conducted in the tillering stage with a dose of 67.5 mg a.i./ha active ingredient at 12 representative locations in China. Appendix A summarizes the specific locations, cultivation time, cultivar types, and sampling duration for each region. In the experiments, a commercial mixture (containing 15% prohexadione-Ca and 15% uniconazole) manufactured by Shanxi Haozhida Biotechnology Co., Ltd. (Shanxi, China) was diluted once and sprayed from these 12 locations. The same volume of tap water was sprayed on the control plots, while each treatment had both a control and a test plot. To avoid cross-contamination, each plot was separated by a buffer zone.

During the experiments, samples (≥2.0 kg) were taken at random intervals (62–112 days) from each of the experimental schemes (3 simultaneous quarters). Healthy rice plants with no pests or diseases were randomly selected at more than 12 points within the experimental area (leaving a 0.5 m margin), and the entire plant, including the panicle and straw, was collected. When sampling the treatment area, control samples were collected simultaneously, and the order of sampling was first the control area, then the treatment area. Two samples were collected from each area each time.

#### 3.2.2. *Citrus reticulata* Blanco

When conducting residue tests of prohexadione-Ca and uniconazole in citrus, we conducted open-field experiments with an active ingredient dose of 300 mg a.i./kg during the summer growing season at 12 representative sites. Appendix A provides details of the location, planting time, cultivar types, and sampling days for each region. To study the effects of prohexadione-Ca and uniconazole in citrus, we used a commercial mixture (manufactured by Shanxi Haozhida Biotechnology Co., Ltd.) containing 15% prohexadione-Ca and 15% uniconazole, diluted it with water, and sprayed it on citrus at 12 locations. Other experimental conditions were consistent with those of the field experiments in rice.

Samples were randomly taken from different parts of the citrus tree (top, bottom, inside, outside, sunny, and shaded sides), excluding samples that were not representative due to pest or disease damage. Two samples were taken each time, and at least twelve citrus fruits (≥2 kg) were collected for each sample at each experimental site. When sampling the treatment area, control area samples were collected at the same time.

### 3.3. Stability Study

The stability of prohexadione, prohexadione-Ca, and uniconazole was studied using samples of rice (brown rice, rice hull, and rice straw) and citrus (whole fruit and pulp) at −18 °C darkness. Approximately 2 g of the crushed brown rice, rice hull, and rice straw, as well as 5 g of the homogenous whole citrus fruit and citrus pulp samples, were each fortified with prohexadione, prohexadione-Ca, and uniconazole at a concentration of 0.5 mg/L. The concentration of prohexadione, prohexadione-Ca, and uniconazole in rice samples were determined after 29, 87, and 179 days of frozen storage, respectively, while the concentration in the citrus samples was detected after 30, 329, and 380 days of frozen storage. Two untreated citrus and rice samples were added at a concentration of 0.5 mg/L for a quality control test before analysis, and an untreated blank and two preserved rice and citrus samples were assayed simultaneously.

### 3.4. Sample Preparation

#### 3.4.1. Prohexadione and Prohexadione-Ca

Extraction: A total of 2 g of crushed rice samples (including brown rice, rice hull, and rice straw) and 5 g of homogenized citrus samples (whole citrus fruit and citrus pulp) were placed in a 50 mL centrifugal pipe. For rice samples, 10 mL of 0.1% formic acid in water (20 mL in rice straw) was added, and for citrus samples, 20 mL of water was poured. The samples were vortexed for 30 min, followed by centrifugation at 8 × 10^3^ g for 5 min. This process was repeated.

SPE procedure: The PAX SPE cartridges were conditioned before loading the supernatant with 5 mL of methanol and 5 mL of pure water. Next, 10 mL of the rice samples’ supernatant and 8 mL of the citrus samples’ supernatant were loaded onto the preconditioned PAX SPE cartridge. The flow velocity was maintained at 3–6 mL/min by adjusting the vacuum level. The cartridges were rinsed with 5 mL of pure water and then washed with 10 mL of methanol. The elution was performed with 5 mL of 10% formic acid in water with methanol (*v*/*v*, 1:9). The eluate was collected and filtered using a 0.22 μm nylon membrane and used for LC/MS-MS analysis.

#### 3.4.2. Uniconazole

Extraction: For the brown rice sample, 5 g was placed in a 50 mL centrifuge tube, while 2 g each of rice hull and straw samples were raised in the centrifugal pipe, then water (5 mL in brown rice, 2 mL in rice hull, and 10 mL in rice straw) and 10 mL of acetonitrile (20 mL acetonitrile to rice straw) were applied sequentially and vortexed for 30 min. Next, 8 g of sodium chloride was given to each tube, and the contents were centrifuged at 4 × 10^3^ g for 5 min. For the citrus fruit sample, 5 g of the homogeneous citrus sample was inserted into a 50 mL centrifugal pipe, and 20 mL ethyl acetate was added. The tube was vortexed for 60 min, followed by centrifugation at 6 × 10^3^ g for 5 min. The resulting fluid was evaporated at 40 °C, and 10 mL of hexane was injected and purified.

SPE procedure: The Florisil (1 g/6 mL) cartridges were conditioned by washing with 5 mL of n-hexane:ethyl acetate (9:1, *v*/*v*), followed by 5 mL of n-hexane. Next, 2 mL of the rice and citrus supernatants were loaded on the preconditioned cartridges, respectively. The flow velocity was controlled at 1–2 mL/min by adjusting the vacuum level. After the sample was loaded, the column was rinsed with 5 mL of n-hexane:ethyl acetate (9:1). Finally, the column was eluted with 5 mL of n-hexane:ethyl acetate (1:1, *v*/*v*), and the eluate was collected and evaporated at 40 °C until nearly dry. The residue was ultrasonicated with 5 mL of methanol, filtered with a 0.22 μm nylon membrane, and used for LC/MS-MS analysis.

### 3.5. LC-MS/MS Conditions

In this work, the LC-MS/MS analysis was performed using an Acquity LC system with a TQ MS Xevo mass spectrometer equipped with an electrospray ionization source (Waters, Milford, MA, USA). The capillary voltage was set to 25,000 V, the desolvation temperature to 600 °C, and the desolvation gas flow (nitrogen) to 1000 L/Hr. The analytical column was maintained at a temperature of 40 °C, and an HSS T3 column (2.1 × 100 mm, 1.8 μm) was used. The mobile phase was a gradient of (A) acetonitrile: (B) water containing 0.1% formic acid. The mobile phase was 0.3 mL/min. The gradient began at 5% A (0–0.5 min), increased to 90% A (0.5–3.0 min), and was held for 1 min (3.0–4.0 min), then decreased to 5% A (4.0–4.5 min) and maintained for a further 1 min (4.0–5.0 min). The injection volume was 1 µL, and detection was carried out by multiple reaction monitoring (MRM) as listed in Table 5.

### 3.6. Method Validation

The method was evaluated by matrix effect, linearity, the limit of detection (LOD), the limit of quantification (LOQ), precision, repeatability, and trueness [41]. The linearity of prohexadione, prohexadione-Ca was measured using six different concentrations (0.002, 0.005, 0.01, 0.05, 0.1 and 0.2 mg/L), and the linearity of uniconazole was measured by applying six gradients (0.0002, 0.0005, 0.001, 0.005, 0.01 and 0.02 mg/L) based on the coefficient of determinations (R^2^) of the calibration curves. The LOQ was used to determine the lowest concentration of the analyte that can be reliably detected and quantified with acceptable performing precision and accuracy. The trueness and precision of the method were determined by including recovery tests at three fortification levels (low, medium, and high) for each analyte in the blank substrate matrices of rice (brown rice, rice hull, and rice straw) and citrus (whole fruit and pulp) samples. The fortification levels were 0.01/0.05, 0.1, and 0.5 mg/kg for rice samples and 0.01/0.05, 0.5, and 1.0 mg/kg for citrus samples. The ME was calculated using the formula ME = [slope (matrix)/slope (solvent) − 1] × 100, where slope (matrix) is the matrix-matched standard calibration curves and slope (solvent) is the solvent standard calibration curves [42].

### 3.7. Dietary Risk Assessment

Prohexadione-Ca and uniconazole long-term dietary risk assessments in citrus and rice were computed using the formulas NEDI = (*STMR_i_* × *E_i_* × *P_i_* × *F_i_*) and RQ% = NEDI/(*ADI* × bw) × 100, in which NEDI is the national estimated daily intake [47]. The average value of the terminal deposit was employed in this investigation, while *STMR_i_* is the median residue level for monitored experiments. The international intake of a specific agricultural commodity or food, expressed in kilograms per day depending on weight, is known as *F_i_*. The symbol RQ stands for the risk quotient. The terms “*E_i_*” and “*P_i_*” denote, respectively, the factor for the consumable component and the processed part of a certain food. The discrepancy was not taken into account in this investigation, and *E_i_* and *P_i_* were both set to 1. Acceptable daily intake (*ADI*) is a toxicological term that represents the amount of a substance that can be ingested daily over a lifetime without any appreciable health risk. It is usually expressed in milligrams of the substance per kilogram of body weight (mg/kg bw). An unacceptable risk is there when *RQ* > 1, and an acceptable risk is present when *RQ* < 1. The risk and *RQ* value have a favorable correlation [48].

## 4. Conclusions

The LC-MS/MS method developed in this study proved to be reliable and efficient for the analysis of prohexadione-Ca and uniconazole residues in *Oryza sativa* L. (brown rice, rice hull, and rice straw) and *Citrus reticulata* Blanco (whole fruit and pulp). The results of the residue and risk assessment indicated that the dietary intake risks for consumers were low, indicating that the use of these two pesticides was safe. In field residue trials conducted at twelve locations in 2020, rice was treated with a dose of 67.5 mg a.i./ha of the active ingredient. The resulting residues of prohexadione-Ca and uniconazole in rice crops at maturity (62–112 days after application) were found to be <0.05 mg/kg. Similarly, in field residue trials conducted at twelve locations in 2019, citrus was applied at active ingredient doses of 300 mg a.i./kg, and the residue at maturity (134–191 days after application) was also <0.05 mg/kg. The study found that the STMR of prohexadione-Ca on rice crops and citrus fruit was 0.05 mg/kg, while the HR was 0.05 mg/kg. For uniconazole, the STMR was <0.01 mg/kg, and the HR was 0.01 mg/kg. Overall, the study provides a reliable LC-MS/MS method for the analysis of prohexadione-Ca and uniconazole residues, which can be useful for future evaluations of the dietary intake risk of these chemicals.

## Figures and Tables

**Figure 1 molecules-28-02611-f001:**
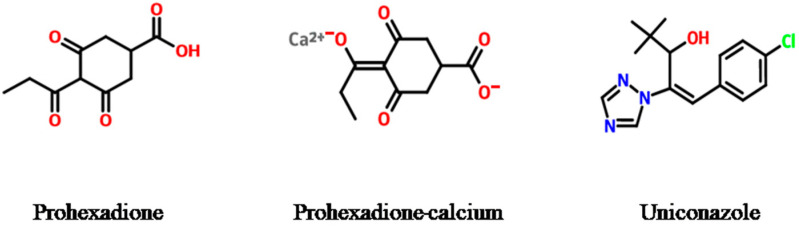
Chemical structure of plant growth regulators (prohexadione, prohexadione-Ca, and uniconazole).

**Figure 2 molecules-28-02611-f002:**
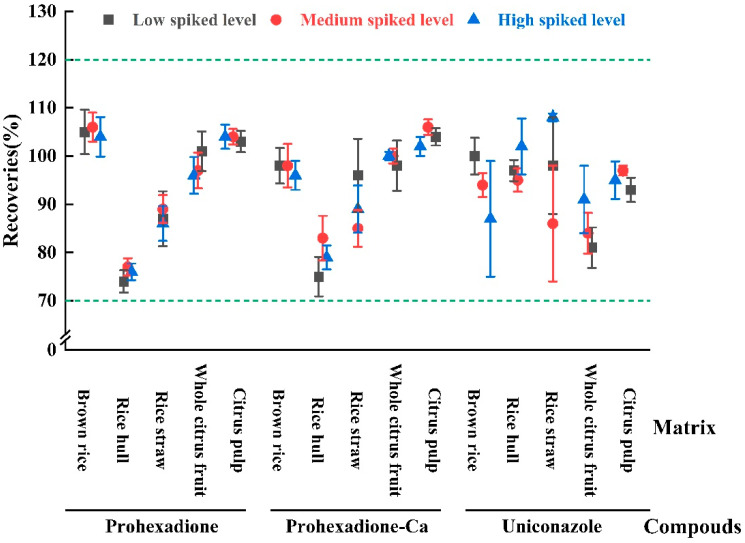
Recoveries of prohexadione, prohexadione-Ca and uniconazole in rice and citrus.

**Figure 3 molecules-28-02611-f003:**
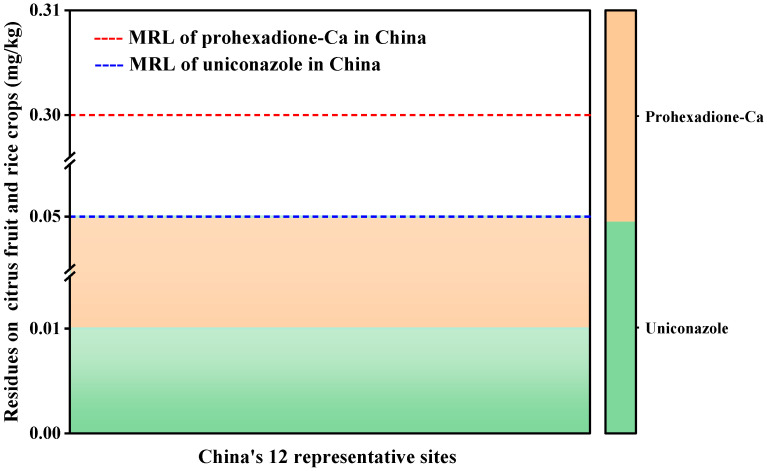
Residues and MRLs of prohexadione-Ca and uniconazole at 12 representative sites.

**Table 1 molecules-28-02611-t001:** Relevant parameters of the matrix-based standard curves.

Compounds	Linearity (mg/L)	Matrix	Regression Equation	Correlation Coefficient (R^2^)	Limit of Detection (LOD, mg/L)	Limit of Quantitation (LOQ, mg/kg)
Prohexadione	0.002–0.2	Brown rice	y = 1085.33x + 79.1864	0.998 8	0.002 0	0.050 0
		Rice hull	y = 1038.4x + 369.114	0.999 8	0.002 0	0.050 0
		Rice straw	y = 878.72x + 58.5053	0.999 9	0.002 0	0.050 0
		Whole citrus fruit	y = 371,672x + 266	0.999 5	0.002 0	0.050 0
		Citrus pulp	y = 377,460x + 351	0.999 9	0.002 0	0.050 0
		Solvent	y = 451,692x − 117	0.999 9	/	/
Uniconazole	0.0002–0.02	Brown rice	y = 1345.25x − 34.3587	0.999 9	0.000 2	0.010 0
		Rice hull	y = 1234.72x − 3.54155	0.999 9	0.000 2	0.010 0
		Rice straw	y = 1156.3x + 1.46754	0.999 9	0.000 2	0.010 0
		Whole citrus fruit	y = 112,837,058x + 2050	1.000 0	0.000 2	0.010 0
		Citrus pulp	y = 125,369,785x − 20,942	0.999 1	0.000 2	0.010 0
		Solvent	y = 83,296,042x + 16,250	0.999 9	/	/

**Table 2 molecules-28-02611-t002:** The residues of prohexadione-Ca and uniconazole on rice and citrus.

Table	Compounds	Matrix	Harvest Interval (Days)	Residue (mg/kg)	STMR ^a^ (mg/kg)	HR ^b^ (mg/kg)
In 2020/ At 12 sites	Prohexadione-Ca	Brown rice	62–112	<0.05 (12)	0.05	0.05
		Rice hull	62–112	<0.05 (12)	0.05	0.05
		Rice straw	62–112	<0.05 (12)	0.05	0.05
	Uniconazole	Brown rice	62–112	<0.01 (12)	0.01	0.01
		Rice hull	62–112	<0.01 (12)	0.01	0.01
		Rice straw	62–112	<0.01 (12)	0.01	0.01
In 2019/ At 12 sites	Prohexadione-Ca	Whole citrus fruit	134–191	<0.05 (12)	0.05	0.05
		Citrus pulp	134–191	<0.05 (12)	0.05	0.05
	Uniconazole	Whole citrus fruit	134–191	<0.01 (12)	0.01	0.01
		Citrus pulp	134–191	<0.01 (12)	0.01	0.01

^a^ STMR = supervised trials median residue. ^b^ HR = the maximum residue value.

**Table 3 molecules-28-02611-t003:** The long-term dietary risk assessment of prohexadione-Ca based on Chinese food diet.

Food Category	Dietary Intake (kg/Person/Day)	MRL ^a^/STMR ^b^ (mg/kg)	Source of Reference Limit	Commodity	NEDI ^c^ (mg)	ADI ^d^ (mg/kg b.w.)	RQ (%)
Rice cereals and rice products	0.239 9	0.05	STMR	Brown rice	0.012 0	ADI × 3 ^e^	
Wheat cereals and wheat products	0.138 5	0.1	EU	Wheat	0.013 9		
Other cereal grains	0.023 3						
Potatos	0.049 5						
Dried beans and their products	0.016						
Dark-colored vegetables	0.091 5						
Light-colored vegetables	0.183 7						
Pickles	0.010 3						
Fruits	0.045 7	0.05	STMR	Mandarin	0.002 3		
Nuts	0.003 9						
Livestocks and poultries	0.079 5						
Milk and milk products	0.026 3						
Egg and egg products	0.023 6						
Fish and fish products	0.030 1						
Oil seeds and oil	0.032 7	1	USA	Peanut	0.032 7		
Animal origin oil and fat	0.008 7						
Sugars and starch	0.004 4						
Salt	0.012						
Soy sauce	0.009						
Total	1.028 6				0.060 8	0.2	0.48%

^a^ MRL, maximum residue limit. ^b^ STMR, supervised trials median residue. ^c^ NEDI, national estimated daily intake. ^d^ ADI, acceptable daily intake. ^e^ 63, the average body weight of a Chinese adult.

**Table 4 molecules-28-02611-t004:** The long-term dietary risk assessment of uniconazole based on Chinese food diet.

Food Category	Dietary Intake (kg/Person/Day)	MRL ^a^/STMR ^b^ (mg/kg)	Source of Reference Limit	Commodity	NEDI ^c^ (mg)	ADI ^d^ (mg/kg b.w.)	RQ (%)
Rice cereals and rice products	0.239 9	0.01	STMR	Brown rice	0.002 4	ADI × 63 ^e^	
Wheat cereals and wheat products	0.138 5	0.05	China	Wheat	0.006 9		
Other cereal grains	0.023 3						
Potatos	0.049 5						
Dried beans and their products	0.016						
Dark-colored vegetables	0.091 5						
Light-colored vegetables	0.183 7						
Pickles	0.010 3						
Fruits	0.045 7	0.01	STMR	Mandarin	0.000 5		
Nuts	0.003 9						
Livestocks and poultries	0.079 5						
Milk and milk products	0.026 3						
Egg and egg products	0.023 6						
Fish and fish products	0.030 1						
Oilseeds and oil	0.032 7	0.05	China	Rapeseed, Peanut	0.001 6		
Animal origin oil and fat	0.008 7						
Sugars and starch	0.004 4						
Salt	0.012						
Soy sauce	0.009						
Total	1.028 6				0.114 2	0.02	1.26%

^a^ MRL, maximum residue limit. ^b^ STMR, supervised trials median residue. ^c^ NEDI, national estimated daily intake. ^d^ ADI, acceptable daily intake. ^e^ 63, the average body weight of a Chinese adult.

**Table 5 molecules-28-02611-t005:** MRM set up on mass spectrometer.

Analyte	Precursor (m/z)	Product (m/z)	Collision Energy (eV)	Retention Time (min)	Ionization Mode
Prohexadione	211.00	123.00 *	14	2.82	ESI(−)
	211.00	167.00	20		
Uniconazole	292.10	70.10 *	24	3.72	ESI(+)
	292.10	125.00	28		

* Quantitative ion.

## Data Availability

Not applicable.

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
