# Peer review of "Residues and Dietary Risk Assessment of Prohexadione-Ca and Uniconazole in Oryza sativa L. and Citrus reticulata Blanco by Liquid Chromatography-Tandem Mass Spectrometry"

_molecules, 2023, doi:10.3390/molecules28062611_

Round 1

Reviewer 1 Report

General comments:

While I am enjoying reading the manuscript, I do want to point out that there are spaces for improvement in manuscript organization. For example, there is no line index across the manuscript which causes lots of troubles for reviewing process as it is hard to referring where the correction/improvement are required.

Some points listed below:

Figures and Tables:

Figure 2: the shapes and error bars of the three levels are heavily overlapped, could the author to improve it a bit as it’s quite hard for readers to distinguish which is which. I think there are two simple ways that can resolve this problem, try to separate each shape a bit to reduce the overlap or set up the transparency to allowing all three shapes are visible.

Figure S1: Please vertically align subplots to enable the easy comparison between samples, eg (a) (should be a1 I assume), (a2), (a3), (a4). It’s not necessary to list both samples, and list only one sample as the representative peak will be enough.

Table 1: Please add the full name of abbreviations like LOD and LOQ. I do notice that these full names were mentioned in the material section, but a full name should be mentioned on its first appear.

Table 2: residue abundance listed on the table 2 are < 0.05 mg/kg and < 0.01 mg/kg for Prohexadione-Ca and Uniconazole, respectively. It’s important to determine that the abundance is below the limitation. However, I remember a standard curve was applied in this study, with that can authors provide the exact abundance of the two target compounds in each of the 12 locations? I believe these would bring more value to the community/industry.

Table 3: I do not see Table 3 neither in main manuscript nor in the supplementary document. Please double check.

Main body:

The manuscript claims that the optimized LC-MS/MS method is simple and has better linearity, trueness and precision when compared to traditional approaches. Have authors done any comparison on the directions like time consumption and cost per sample? Such a section of discussion on these items will not clarify your claim but also help researchers and/or companies to determine which method they should apply for their purpose.

Subsection 2.3: samples were collected at the range of 62-112 days after application for rice and that of 134-191 days after application for citrus. This is a big range of sampling time, and I assume there were three samplings (one control, two treatments in a random intervals mainner). Can you please specify when/how exactly these samples were collected? And why the sampling range is this big? For the majority circumstances, the minimal/standard sample size would be 3 replicates and you might wanted to clarify a bit why there are only one (control) and two (treatments) reps are included in this study.

Subsection 2.5.1: “The MRL for pro-hexadione-Ca in citrus fruits published by America is 3.0 mg/kg, whereas the MRL set by China for uniconazole is 0.3 mg/kg”, these are two different MRL in two countries and are not comparable, I think you may want to compare the same MRL in different countries. The manuscript has mentioned that the Prohexadione-Ca and Uniconazole MRL for rice and its products are 0.05 and 0.3mg/kg in China. Is there any country has the even lower MRL, if so how’s your result compared to that? I believe a brief discussions on this perspective would be useful.

Subsection 3.3: the measurement dates that were chosen seems a bit odd to me (eg. 30, 329, and 308 for Prohexadione-Ca). As shown in the Table S1, the compounds are quite stable and there is no significant degradation were occurred over the time course selected. In which case, why the authors choose these two dates and the gaps between dates are dramatically different?

Subsection 3.5: (the first paragraph in page 10), I think the LC-MS/MS gradient description was duplicated.

Reviewer 2 Report

Recommendation: Publish after minor revision. 

Comments:

This high-quality research article investigates a range of factors to be considered for the analysis of prohexadione–Ca and uniconazole including linearity, recoveries, precision, sensitivity and repeatability, which build a standard workflow to evaluate their dietary intake risk.  The experiment design is straightforward and the text is clear and well-written. While this work has already been organized well, I would recommend a minor revision and the following points should be addressed.

1. P.2 Figure 1, the structure of prohexadione–Ca, “Ca 2+” is blurred and please redraw the structure.

2. P.3 “so the extraction agent water and formic acid water oscillation”, please consider adding the ratio of formic acid/water. Readers may be interested in the ratio if they don’t look over Section 3.4.

3. P.3 “formic acid water-methanol”, please consider adding the ratio of formic acid water/methanol. 

4. P.5 “There is a representative blank, solvent, calibration curves, and added and tested for the various matrix chromatograms are available in Figures S1 and S2.”, this sentence sounds cryptic. Please rephrase it or consider adding (a), (b), (c)…like Figure S1 and Figure S2 did.

5. P.10 “The mobile phase gradient started at 5% A (0–0.5 min), increased to 90% A (0.5– 3.0 min), and held on for 1 min (3.0–4.0 min) and decreased at 5% A (4.0–4.5 min) and hold on 1 min (4.0–5.0 min). An injection volume of 1 mL was used, and the method of checkout was MRM (multiple reaction monitoring, MRM) listed in Table 6.”, please delete the same content.

6. P.11 Figure S1 and Figure S2, is it possible to update the figure with a higher resolution? Probably can split the single picture to several parts or use a software to create graphs instead of using screenshots.
